# An integrated model of threshold-based scaling and fractional admission controlling to improve resource utilization efficiency in 5G core networks

**Ly Cuong Hoa**[1,2], **Thanh Chuong Dang**[2]*, **Viet Minh Nhat Vo**[3]

**1** College of Information and Communication Technology, Can Tho University, Can Tho City, Vietnam,
**2** University of Sciences, Hue University, Hue City, Vietnam, **3** Hue University, Hue City, Vietnam

\* dtchuong@hueuni.edu.vn

**Data availability statement:** All relevant data are within the manuscript and its Supporting Information files.

## Abstract

User Plane Function (UPF) is considered a bridge between User Equipment (UE) and Data Networks (DN) in the 5G core network. A UPF instance can manage multiple Packet Data Unit (PDU) sessions, and there are usually various UPF instances deployed to serve PDU session requests. One requirement is utilizing system resources effectively while ensuring stable system performance. Specifically, the need to optimize unused UPF instances to reduce system costs. The paper proposes a fractional admission controlling (FAC) mechanism and integrates it with a Markov chain-based analytical model for threshold-based scaling for UPF instances (called TSUPF-FAC), in which two additional thresholds are added to control UPF instances globally in order to optimize resource utilization. A threshold-based scaling and fractional admission controlling (TS-FAC) algorithm is developed and implemented in Kubernetes-based Open5GS. The simulation results show a similarity between the analytical and experimental results, in which the analytical model helps to determine the admission thresholds for the best performance of TSUPF-FAC, as measured by metrics such as the number of idle UPF instances and system utilization.

## Introduction

The 5G network is a new generation of mobile networks capable of meeting the many requirements of vertical industries. In the 5G network with Service-Based Architecture, control and user planes are separated, which provides flexibility in allocating UPF instances for new 5G applications and customer-specific edge services. UPF instances are responsible for connecting Radio Access Networks (RAN) to DNs via Access and Mobility Management Functions (AMF). A single UPF instance can manage multiple PDU sessions, while a PDU session can only be managed by one UPF instance. PDU sessions provide an end-to-end user plane connections between UEs and DNs [1].

**Funding:** This work was supported by the Ministry of Education and Training (Vietnam) for the development of Science and Technology under grant number B2023-DHH-17. The funders had no role in study design, data collection and analysis, decision to publish, or preparation of the manuscript.

**Competing interests:** The authors have declared that no competing interests exist.

Establishing a PDU session on an UPF instance is illustrated in Fig 1. First, an UE must establish an UPF instance by registering to access the 5G infrastructure. The AMF requests a Session Management Function (SMF) to create a session management context to manage the UE's PDU session. The SMF then selects an UPF instance to serve the UE. In order to create UE-specific QoS flows, a PDU session request initiated by the UE is sent to an UPF instance. The SMF schedules static and dynamic rules for sessions and then establishes session-related rules and policies for the UPF instance. Therefore, each PDU session is managed by an UPF instance, and the operation of UPF instances are performed by the Operation, Administration, and Management. Typically, several UPF instances are deployed to serve PDU session requests. However, since the number of UPF instances is limited, deploying more UPF instances than needed will result in waste. Therefore, a mechanism to control resources while scaling UPF instances is needed.

The paper proposes a FAC mechanism and integrates it with TSUPF (called TSUPF-FAC), in which two different thresholds are added to control UPF instances globally. A Markov chain-based analytical model (called the Queueing model) has also been improved to evaluate the effectiveness of the new integration model. A TS-FAC algorithm has been developed and implemented in Kubernetes-based Open5GS.

The main contributions of the article include:

- Proposing a FAC mechanism to efficiently respond to PDU session requests;
- Improving the model of TSUPF proposed in [2] by integrating a FAC mechanism to scale UPF instances effectively. The improved model is called the Threshold-based Scaling and Fractional Admission Controlling for UPF instances (TSUPF-FAC);
- Building a Queueing model for TSUPF-FAC (Q- TSUPF-FAC) to evaluate the impact of control thresholds on the performance of TSUPF-FAC; and
- Developping a TS-FAC algorithm and implementing simulation on Kubernetes with Open5GS to evaluate the performance of TSUPF-FAC experimentally.

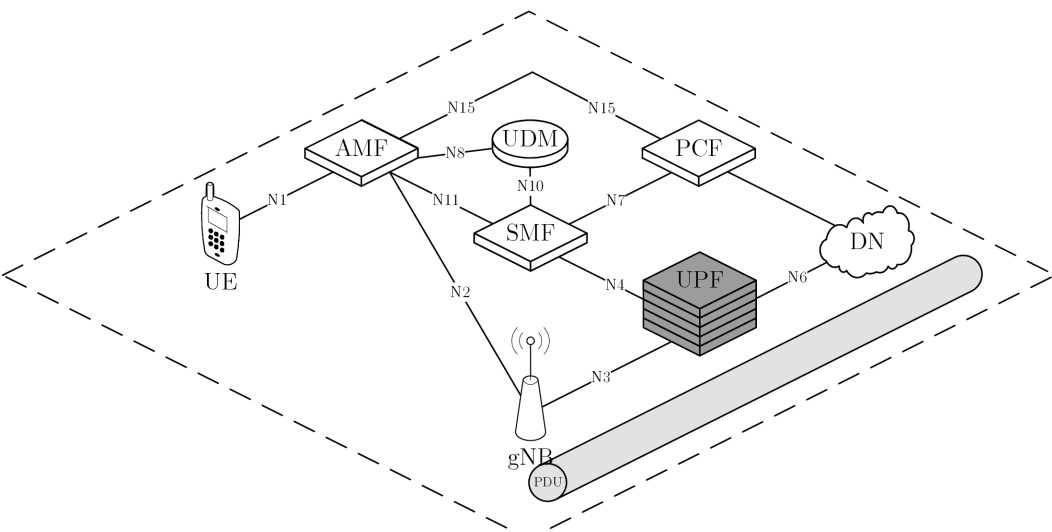

**Fig 1. The model of establishing PDU sessions in an UPF instance in the 5G core network.** Functions of gNB, AMF, SMF, and UPF in PDU session establishment between UE and the data network.

The following sections of the article are organized as follows. Sect 'Literature reviews' introduces related works. The TSUPF-FAC model and queueing-based performance analysis are presented in Sect 'Thresholds-based scaling for UPF instances with fractional admission controlling'. Experimental implementation and result analysis are described in Sect 'Results and discussions'. The rest is the conclusion and future research.

## Literature reviews

In 5G core networks, network functions can be dynamically scaled in/out to adjust the capacity of network components (e.g., UPF instances, Virtualized Network Functions (VNF), network slices). The process of scaling out instances is to increase available resource capacity, while the process of scaling in instances is to reduce operational costs. However, scaling-in/out issues in 5G networks differ from those in traditional cloud computing [3]. 5G network functions must deploy multiple instances simultaneously and more frequently than traditional cloud computing. Both the number and frequency of deployments impact cost-effectiveness significantly. One such system, called Telco-Cloud, was introduced by [4], which aims to deploy VNFs with the ability to handle an enormous number of requests.

There have been some studies on scaling with different resource types. Herrera and Moltó [5] studied the impact of a bio-based approach on automated container orchestration platforms. They found a relationship between bio-based approaches and the scalability of containers. The scaling method is also considered for improvement in some actual cases. Taherizadeh et al. [6] proposed an auto-scaling method based on a dynamic multi-level model, where the thresholds change automatically and the application scope is not only limited to the network infrastructure but can also be applied to application monitoring data. Guo et al. [7] developed a technique for packaging virtual machines to physical machines (VM-to-PM) and how to scale virtual machines to meet resource requirements. The Monitor-Analyze-Plan-Execute (MAPE) method was also introduced by Nguyen et al. [8] to manage the auto-scaling of UPF instances.

There have been several proposals for Network Functions Virtualization (NFV) and VNF. Specifically, the NFV auto-scaling algorithm proposed by Ren et al. [9,10] considered the trade-off between performance and operating costs. VNF is also enabled or disabled depending on the required capacity. Kumar et al. [11] proposed a new method for scaling in and out of VNFs and thereby discovered techniques for allocating and revoking UPF instances based on the Linux kernel. Ren et al. [12] introduced the VNF auto-scaling algorithm to balance high performance and low operating costs. Tang et al. [13] introduced dynamic VNF scaling and deployment in data center networks. A system called ScalFlux was introduced by Liu et al. [14] to reduce latency and achieve optimal performance through VNF traffic monitoring. Adamuz-Hinojosa et al. [15] proposed a scaling process according to European Telecommunications Standards Institute (ETSI) standards through interaction and information exchange between functional blocks in the NFV framework. The problem of optimizing Service Function Chain design and VNF placement to minimize resource costs considering VNF dependencies and traffic scale was addressed in the research of Zeng et al. [16]. Chen et al. [17] proposed HyScaler as a hybrid system for scaling VNFs deployed on an open-source NFV platform. Another tool called Tacker, based on the OpenStack open-source cloud computing platform, was also used by Sales et al. [18] to implement autoscaling functions in NFV. Leyva-Pupo et al. [19,20] raised the issue of dynamic UPF position reconfiguration due to user mobility. They proposed an Integer Linear Programming (ILP) model to reduce the cost and scheduling mechanism for re-computation time.

System performance enhancement can be achieved by Guard Channel (GC) and Fractional Guard Channel (FGC) mechanisms. Cruz-Perez and Ortigoza-Guerrero [21] overviewed call admission control mechanisms to ensure QoS in mobile networks. The authors [21] proposed the FGC mechanism to improve the original GC. Chuong et al. [22] consider the retrial queueing model with the FGC mechanism in cellular networks, thereby analyzing, evaluating, and comparing four mechanisms of Limited FGC, Uniform FGC, Limited Average FGC, and Quasi Uniform FGC.

Barrachina-Muñoz et al. [23] have developed and evaluated a 5G framework based on network functions encapsulated within Kubernetes clusters. Similarly, using the Kubernetes platform, Simone et al. [24] employed queueing models for AMF, SMF, and UPF nodes to assess system performance related to latency. Accordingly, our model presented in this paper will also be modeled based on Kubernetes.

The Queueing model used to analyze the resource scaling (e.g., UPF instances) has also attracted the attention of some researchers. Accordingly, Hsieh et al. [3] applied the queueing model to analyze the scaling of resource blocks corresponding to network slices. Rotter and Do [2] proposed the Queueing model of Threshold-based Scaling for UPF instances (Q-TSUPF) to analyze the scaling of UPF instances when the number of sessions being served reaches thresholds $T_1$ or $T_2$. However, this threshold pair is considered locally for each UPF to scale in or out when the number of PDU sessions reaches them. This approach clearly does not consider the entire system's remaining UPF instances when scaling. In the case of a sudden increase in the number of PDU session requests, while the number of UPF instances is almost exhausted, it may be impossible to satisfy PDU session requests, and as a result, the service efficiency of the system is seriously degraded. Controlling the admission of PDU session requests is therefore necessary to improve the efficiency of resource use, and maintain stable system performance.

This paper will propose an improved model of TSUPF, in which a fractional admission controlling mechanism is integrated to control the admission of incoming PDU session requests, thereby more effectively managing the deployment or termination of UPF instances. A Queueing model for analyzing TSUPF-FAC is also developed, and implementating TSUPF-FAC on Kubernetes with Open5GS is also deployed. The following section will describe our contributions in detail.

## Thresholds-based scaling for UPF instances with fractional admission controlling

### Problem description

The UPF plays a crucial role in 5G networks to realize the transformation of low latency and high throughput. The UPF is deployed as software and packaged in virtual machines or image containers. Service providers launch UPF instances in their cloud infrastructure to serve customers. In 5G networks, there is significant variation in PDU sessions generated by subscriber devices. In order to ensure quality of service (QoS), each UPF instance typically handles only a limited number of PDU sessions. New UPF instances can be deployed when there are more PDU session requests, and idle UPF instances are terminated when the number of PDU sessions decreases. In other words, improved scaling algorithms are required to manage UPF instances efficiently.

In [2], a Queueing model with the scaling-in ($T_1$) and scaling-out ($T_2$) thresholds is introduced to deploy and terminate an UPF instance when the number of in-serve PDU sessions are $N_j - T_1 - 1$ and $N_j - T_2 + 1$, respectively. A two-dimensional Markov chain with continuous

time $(I(t), J(t)), t \geq 0$, where $I(t)$ is the number of in-serve PDU sessions and $J(t)$ is the number of UPF instances deployed at time $t$, is analysed. The arrival rate of PDU session requests ($\lambda$) is assumed to have a Poisson distribution, and the service time of PDU sessions ($1/\mu$) is assumed to have an exponential distribution. In fact, the assumption of Poisson-distributed session arrivals in 5G networks has been introduced in [2,12,25,26]. Poisson distribution is often used to model events such as incoming calls, network connection requests, or data sessions (e.g., PDU sessions in 5G networks). Non-Poisson-distributed arrivals also exist in 5G networks, but this paper only focuses on Poisson-distributed arrivals.

An UPF instance ($j$) only is deployed, with the state transited from $j$ to $j + 1$, $M \leq j < L$, if the number of in-serve PDU sessions in-serve ($i$) increases from $N_j + C - T_2$ to $N_j + C - T_2 + 1$, denoted by $(N_j + C - T_2, j) \rightarrow (N_j + C - T_2 + 1, j + 1)$. Conversely, an UPF instance ($j$) is terminated, with the state transited from $j$ to $j–1$, $M < j \leq L$, if the number of in-serve PDU sessions in service ($i$) decreases from $N_j - C - T_1$ to $N_j - C - T_1 - 1$, denoted by $(N_j - C - T_1, j) \rightarrow (N_j - C - T_1 - 1, j - 1)$. The threshold-based scaling problem can be illustrated in Fig 2.

The transition process of a state ($i,j$) is carried out as in [2]:

- From ($i,j$) to ($i + 1, j$): When a new PDU session request arrives and the system can serve it, a new UPF instance is not deployed if:
    - either $0 \leq i \leq N_M - T_1 - 2, j = M$;
    - or $N_j - T_2 < i \leq N_j - C - T_1 - 2, M < j < L$;
    - or $N_L - T_2 < i \leq N_L - 1, j = L$.

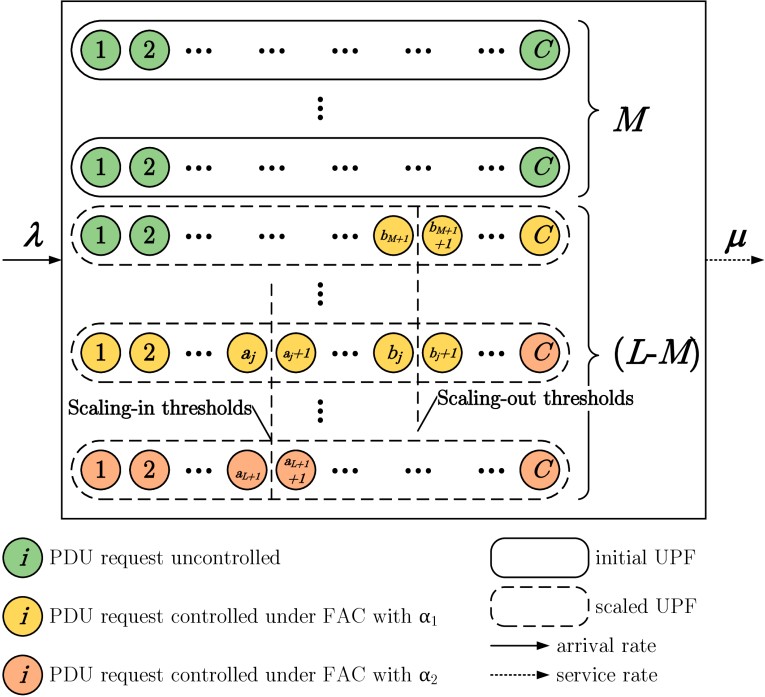

**Fig 2. Operation of TS-FAC for UPF instances.**

- From $(i,j)$ to $(i+1, j+1)$: When a new PDU session request arrives and the system can serve it, an UPF instance is deployed if $i = N_j - T_1 - 1, M \le j < L$.
- From $(i,j)$ to $(i-1,j)$: When a PDU session departs, a deployed UPF instance is not terminated if:
  - either $0 \le i \le N_M - T_1, j = M$;
  - or $N_j - T_2 + 2 < i \le N_j - T_1 - 2, M < j < L$;
  - or $N_L - T_2 + 2 < i \le N_L, j = L$.
- From $(i,j)$ to $(i-1, j-1)$: When a PDU session departs, a deployed UPF instance is terminated if $i \le N_j - T_2 + 1, M < j \le L$.

Note that it is necessary to distinguish two cases: $i' = N_j + C - T_2$ and $i'' = N_j - C - T_1$. This paper only considers the case $i' \ge i''$, that is $N_j + C - T_2 \ge N_j - C - T_1$ or $T_2 - T_1 \le 2C$. Other cases have been demonstrated in [2].

## FAC mechanism

To improve the effectiveness of Q-TSUPF [2], we add two thresholds, $H_1$ and $H_2$ ($1 \le H_1 \le H_2$), to minimize the number of redundant deployed UPF instances while ensuring utilization remains unchanged. Two probabilities of $\alpha_1$ and $\alpha_2$ ($0 < \alpha_2 \le \alpha_1 \le 1$) are also included to control the conditions under which two thresholds of $H_1$ and $H_2$ are applied. $H_1$ and $H_2$ are independent of the deployment threshold $T_1$ and the termination threshold $T_2$. The conditions applying $H_1$ and $H_2$ are as in the FGC mechanism [21]. These two thresholds are intended to gradually reduce the number of deployed UPF instances if the number of in-serve PDU sessions reaches the strict $H_1$ threshold and then the more stringent $H_2$ threshold (Table 1). These thresholds help avoid over-deploying which causes wast of resources) or under-deploying, which leads to degraded performance).

The proposed FAC mechanism in this paper is inspired by the FGC mechanism [21], which is to limit new PDU session requests using two thresholds $H_1$ and $H_2$. Based on the number of in-serve PDU sessions, the FAC mechanism establishes the thresholds $H_1$ and $H_2$ with probabilities $\alpha_1$ and $\alpha_2$, respectively. FAC can be consided as a general case of FGC when not limited by thresholds. However, certain systems may limit the thresholds to a specific range. The FAC mechanism is defined as follows.

**Table 1. Notations used in the FAC mechanism.**

| Symbols | Descriptions |
|---|---|
| $C$ | The maximum number of PDU sessions that a single UPF instance can serve |
| $M$ | The number of initial UPF instances |
| $L$ | The number of maximum UPF instances |
| $T_1$ | The scaling-out decision threshold |
| $T_2$ | The scaling-in decision threshold |
| $\lambda$ | The rate of session arrivals |
| $\mu$ | The reciprocal of the average holding time of sessions |
| $N_j = C \times j$ | The maximum number of PDU sessions that $j$ UPF instances can serve ($M \le j \le L$) |
| $p_{i,j}$ | The steady state probability of state $(i,j)$ |
| $H_1$ | The strict threshold of FAC |
| $H_2$ | The more stringent threshold of FAC |
| $\alpha_1$ | The probability of triggering $H_1$ |
| $\alpha_2$ | The probability of triggering $H_2$ |
| $\beta_{i,j}$ | The control probability of state $(i,j)$ |

**Definition FAC.** *Fractional admission control is a mechanism that deploys UPF instances based on the number of in-serve PDU requests to segment by establishing two thresholds $H_1$ and $H_2$ corresponding to two probabilities $\alpha_1$ and $\alpha_2$, respectively. These parameters are used to determine the probability $\beta_{i,j}$ of system state (i,j) in a two-dimensional Markov chain $\{(I(t), J(t)), t \geq 0\}$.*

## TS-FAC algorithm

Based on **Definition FAC**, we further refine the thresholds-based scaling model [2] by introducing two thresholds and the probabilities of applying them to limit deployed resources and ensure system performance, as presented in **Algorithm TS-FAC**. The purpose of admission controlling is to prevent uncontrolled deploying, system underutilization, and waste of resource.

**Algorithm TS-FAC:** Threshold-based Scaling and Fractional Admission Controlling

```
Input: M, L, C, λ, μ, T₁, T₂, H₁, H₂, α₁, α₂.
Output: J(t)
```

1: $I(t) \leftarrow 0$
2: $J(t) \leftarrow M$
3: **while** $I(t) \leq L \times C$ **do**
4:     $N_{J(t)} \leftarrow J(t) \times C$
5:     $\beta_{I(t),J(t)} \leftarrow 1$
6:     **if** $H_1 < I(t) \leq H_2$ **then**
7:         $\beta_{I(t),J(t)} \leftarrow \alpha_1$
8:     **end if**
9:     **if** $H_2 < I(t)$ **then**
10:         $\beta_{I(t),J(t)} \leftarrow \alpha_2$
11:     **end if**
12:     $\lambda_{I(t),J(t)} \leftarrow \lambda \times \beta_{I(t),J(t)}$  //$\lambda_{I(t),J(t)}$ `are arrival rates at states` $(I(t),J(t))$
13:     **if** `a new UE requests a PDU session setup` **then**
14:         $I(t) \xleftarrow{\lambda_{I(t),J(t)}} I(t) + 1$
15:         **if** $I(t) = N_{J(t)} - T_1$ **and** $J(t) < L$ **then**
16:             **return** $J(t) \leftarrow J(t) + 1$
17:         **end if**
18:     **end if**
19:     **if** `an UE in the system departs` **then**
20:         $I(t) \leftarrow I(t) - 1$
21:         **if** $I(t) = N_{J(t)} - T_2$ **and** $J(t) \geq M + 1$ **then**
22:             **return** $J(t) \leftarrow J(t) - 1$
23:         **end if**
24:     **end if**
25: **end while**

By applying the addition and multiplication rules, as well as the algorithmic complexity calculation methods as in [27], we deduce that the algorithm's complexity is $O(C \times L)$. From the algorithmic complexity, we can see that it depends linearly on $C$ and $L$, and is independent of traffic load.

**Algorithm TS-FAC** manages UPF instances dynamically based on predefined thresholds which helps maintain better performance without over-deploying or under-deploying UPF instances and waste of resource.

One improvement of **Algorithm TS-FAC** is that it considers the probability control value for the states $(i,j)$ as $\beta_{i,j}$ with initial default values of 1. The algorithm then compares the states $(i,j)$, and if $H_1 < i \leq H_2$ then $\beta_{i,j}$ is $\alpha_1$, and if $i > H_2$ then $\beta_{i,j}$ is $\alpha_2$.

## Operating the Queing model of TSUPF-FAC (Q-TSUPF-FAC)

TSUPF-FAC performs scaling of UPF instanses based on data related to the PDU session at the AMF and UPF (Fig 1). The operation of TSUPF-FAC is depicted in Fig 2, where scaling and FAC are performed by **Algorithm TS-FAC**.

Accordingly, the system has at most $L$ deployed UPF instances and each deployed UPF instance has at most $C$ PDU sessions in service. Thus, there is a maximum of $C \times L$ PDU sessions operating in the system. As shown in Fig 2, the uncontrolled cells with green color have the probability $\beta_{i,j} = 1$. The yellow cells controlled at the threshold $H_1$ with the probability $\beta_{i,j} = \alpha_1$. The red cells controlled at the threshold $H_2$ with the probability $\beta_{i,j} = \alpha_2$. Q-TSUPF-FAC is developed from Q-TSUPF in [2]. Specifically, when the number of PDU session requests in the system reaches $N_j - T_1 - 1$, labeled as $a_j$ in Fig 2, the system deploys a new UPF instance. Similarly, if the number of PDU session requests decreases to $N_j - T_2 + 1$, labeled as $b_j + 1$ in Fig 2, the system will terminate one UPF instance. According to **Definition FAC**, the lines 5-12 of **Algorithm TS-FAC** can be described more clearly as follows:

- In the case $H_1 < H_2$,
    - If $H_2 \leq C \times L$ and $i \leq H_1$ then $\beta_{i,j} = 1$;
    - If $H_2 \leq C \times L$ and $H_1 < i \leq H_2$ then $\beta_{i,j} = \alpha_1$;
    - If $H_2 < C \times L$ and $H_2 < i$ then $\beta_{i,j} = \alpha_2$.
- In the case $H_1 = H_2$,
    - If $i \leq H_1$ then $\beta_{i,j} = 1$;
    - If $i > H_1$ then $\beta_{i,j} = \alpha_2$.

For exceptional cases, if $H_1 = H_2 = C \times L$ or $\alpha_1 = \alpha_2 = 1$ then $\beta_{i,j} = 1$, Q-TSUPF-FAC becomes to Q-TSUPF in [2].

## System state equilibrium equations

From Q-TSUPF-FAC in Fig 2, state equilibrium equations and state transition schemes are as the following equations, where equilibrium probabilities of the two-dimensional Continuous Time Markov Chain $(I(t), J(t)), t \geq 0$ is denoted as

$$p_{i,j} = \lim_{t \to +\infty} P(I(t) = i, J(t) = j), (i,j) \in \mathcal{S},$$

Where $\mathcal{S}$ is the set of states of the system: $\mathcal{S} = \{(i,M) : 0 \leq i \leq N_M - T_1 - 1\} \cup \{(i,j) : N_j - T_2 + 1 \leq i \leq N_j - T_1 - 1, M < j < L\} \cup \{(i,L) : N_L - T_2 + 1 \leq i \leq N_L\}$. The cardinality of the set $\mathcal{S}$ is determined as $|\mathcal{S}|$:

$$|\mathcal{S}| = N_M - T_1 + T_2 + (T_2 - T_1 - 1)(L - M - 1) \tag{1}$$

We have state transition equations:

$$p_{i,L} i\mu = p_{i-1,L} \lambda \beta_{i-1,L}, (N_L - C - T_1 < i \leq N_L) \tag{2}$$

$$p_{N_j-T_2+1,j}[\lambda\beta_{N_j-T_2+1,j} + (N_j - T_2 + 1)\mu]$$
$$= p_{N_j-T_2+2,j}(N_j - T_2 + 2)\mu, (M < j \leq L) \tag{3}$$

$$p_{N_j-C-T_1,j}[\lambda\beta_{N_j-C-T_1,j} + (N_j - C - T_1)\mu]$$
$$= p_{N_j-C-T_1-1,j-1}\lambda\beta_{N_j-C-T_1-1,j-1} + p_{N_j-C-T_1-1,j}\lambda\beta_{N_j-C-T_1-1,j}$$
$$+ p_{N_j-C-T_1+1,j}(N_j - C - T_1 + 1)\mu, (M \leq j < L) \tag{4}$$

$$p_{N_j+C-T_2,j}[\lambda\beta_{N_j+C-T_2,j} + (N_j + C - T_2)\mu]$$
$$= p_{N_j+C-T_2,j}\lambda\beta_{N_j+C-T_2,j} + p_{N_j+C-T_2+1,j}(N_j + C - T_2 + 1)\mu$$
$$+ p_{N_j+C-T_2+1,j+1}(N_j + C - T_2 + 1)\mu, (M < j \leq L) \tag{5}$$

$$p_{N_j-T_1-1,j}[\lambda\beta_{N_j-T_1-1,j} + (N_j - T_1 - 1)\mu]$$
$$= p_{N_j-T_1-2,j}\lambda\beta_{N_j-T_1-2,j}, (M \leq j < L) \tag{6}$$

$$p_{i,j}(\lambda\beta_{i,j} + i\mu) = p_{i-1,j}\lambda\beta_{i-1,j} + p_{i+1,j}(i + 1)\mu,$$
$$i \notin \{N_j - T_2 + 1 \mid M \leq j < L\} \cup \{N_j - C - T_1 \mid M \leq j < L\}$$
$$\cup \{N_j + C - T_2 \mid M \leq j < L\} \cup \{N_j - T_1 - 1 \mid M \leq j < L\} \tag{7}$$

The state transition diagrams of Q-TSUPF-FAC corresponding to the Eqs (2)–(7) are shown in Figs 3 to Fig 8.

The Eqs (2)–(7) are solved using a system of linear regression equations with a normalization condition by setting $p'_{i,j} = p_{i,j}/p_{N_L,L}$. In this case, $p'_{N_L,L} = 1$. At this point, the system of the Eqs (2)–(7) can be solved by backtracking [28] since there is a defined value of $p'_{i,j}$. After determining the $p'_{i,j}$ base on the normalization condition in the Eq (8),

$$\sum_{(i,j)\in\mathcal{S}} p_{i,j} = 1 \tag{8}$$

We can calculate:

$$p_{N_L,L} = \frac{1}{\sum_{(i,j)\in\mathcal{S}} p'_{i,j}} \tag{9}$$

We then derive the probabilities $p_{i,j}$ from $p_{i,j} = p'_{i,j} \times p_{N_L,L}$.

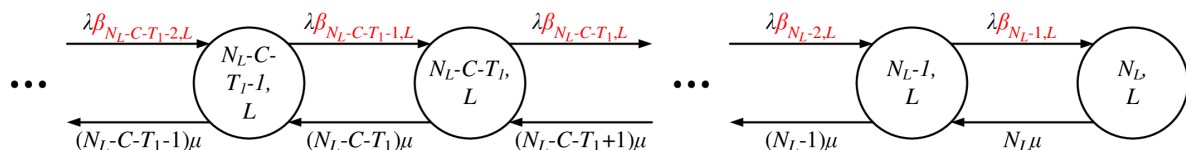

**Fig 3. State transition subdiagram of Q-TSUPF-FAC for the Eq (2).**

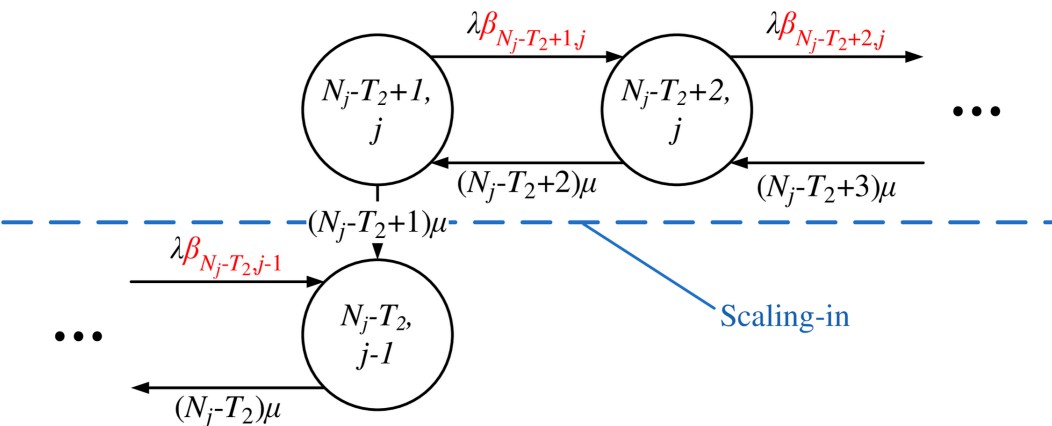

**Fig 4. State transition subdiagram of Q-TSUPF-FAC for the Eq (3).**

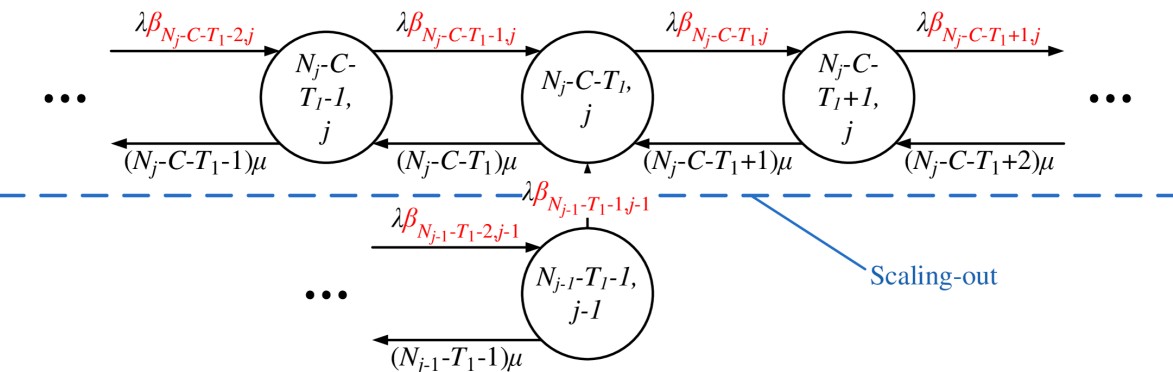

**Fig 5. State transition subdiagram of Q-TSUPF-FAC for the Eq (4).**

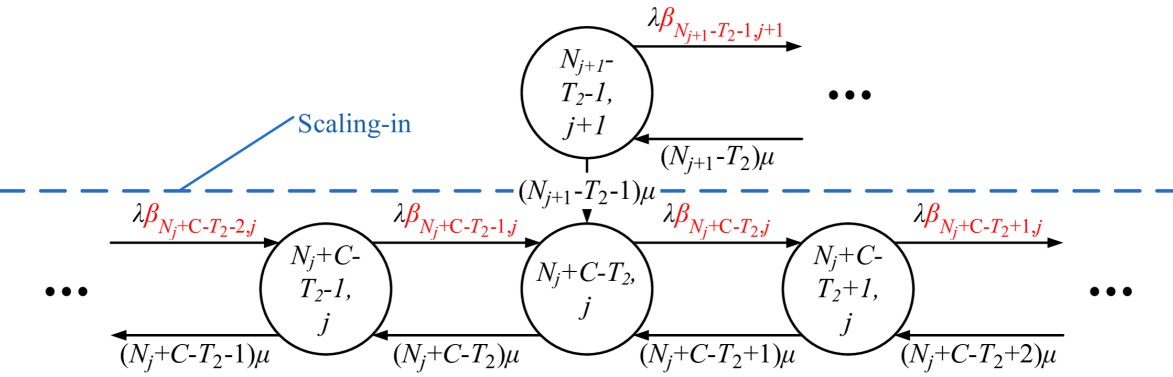

**Fig 6. State transition subdiagram of Q-TSUPF-FAC for the Eq (5).**

### Performance evaluation metrics

The performance metrics of Q-TSUPF-FAC are similar to those of Q-TSUPF [2]. However, Q-TSUPF-FAC adds the probability $\beta_{i,j}$ to achieve the minimal number of idle UPF instances.

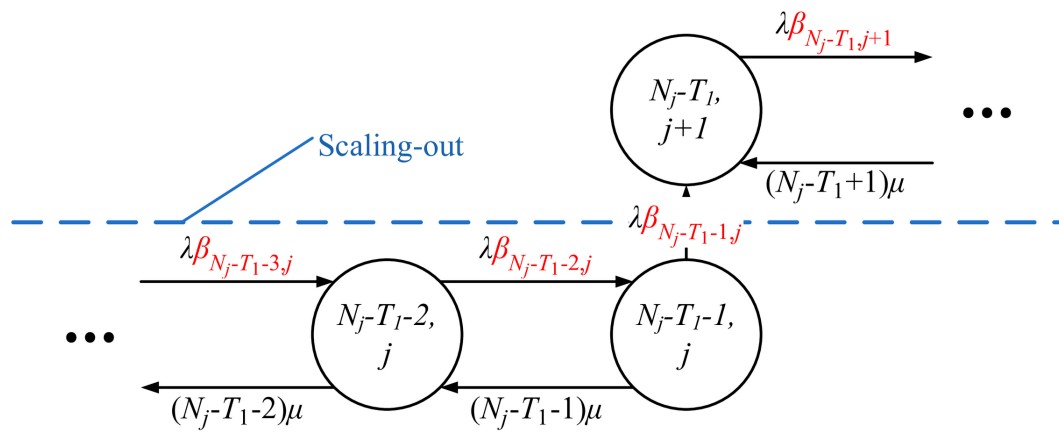

**Fig 7. State transition subdiagram of Q-TSUPF-FAC for the Eq (6).**

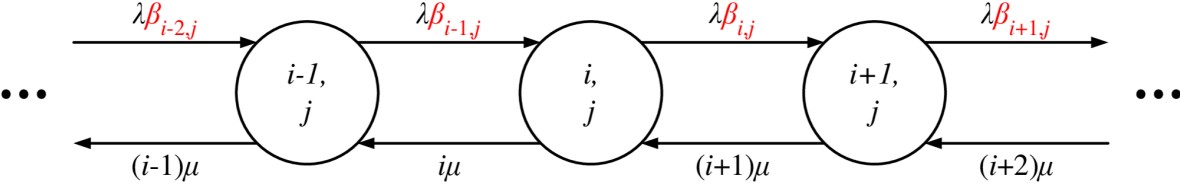

**Fig 8. State transition subdiagram of Q-TSUPF-FAC for the Eq (7).**

- The average number of deployed UPF instances includes the busy and idle UPF instances.

$$V_d = \sum_{(i,j)\in\mathcal{S}} j p_{i,j} \tag{10}$$

- The average number of busy UPF instances includes the deployed and used UPF instances.

$$V_b = \sum_{(i,j)\in\mathcal{S}} \left\lceil \frac{i}{C} \right\rceil p_{i,j} \tag{11}$$

- The average number of UPF instances deployed but idle ($V_{id}$) is for insurance purposes. However, sometimes, the number of idle UPF instances exceeds the required insurance, which is called redundant UPF instances. The redundancy leads to a waste of resources and increased management costs. Therefore, Q-TSUPF-FAC aims to minimize the number of redundant UPF instances.

$$V_{id} = V_d - V_b \tag{12}$$

- Utilization is the ratio of used resources to allocated resources.

$$U = \sum_{(i,j)\in\mathcal{S}} \frac{i}{jC} p_{i,j} \tag{13}$$

The probability $\beta_{i,j}$ in Q-TSUPF-FAC directly affects the equilibrium probability $p_{i,j}$, which improves the performance parameters of Q-TSUPF-FAC compared to Q-TSUPF in [2].

To evaluate the effectiveness of the TSUPF-FAC model, we assess both the $V_{id}$ and $U$ metrics in comparison with the model presented in [2]. The subsequent analysis will show that the $V_{id}$ value in our model is smaller than the $V_{id}$ value in [2], while the $U$ value in Q-TSUPF-FAC is larger than the $U$ value in [2]. This indicates that the Q-TSUPF-FAC model has deployed a sufficient number of idle UPF instances, thereby decreasing redundant UPF instances and helping to reduce system costs. The next section will clarify the advantages of our proposed model compared to a specific scenario identified in [2].

## Implementing of TS-FAC on Kubernetes with Open5GS

To evaluate the proposed Q-TSUPF-FAC model, we model the 5G core network with UPF instances, as depicted in Fig 9 [23,24].

The simulation model in Fig 9 depicts the architecture of the 5G core network using the Open5GS simulator integrated within Kubernetes, encompassing various functional blocks [23,24]:

- UERANSIM block: models the open-source simulation of UE and RAN.
- Open5GS block: implements the core infrastructure of the 5G network in compliance with 3GPP Release 17 [29] and serves as the deployment area for the various versions of UPF.

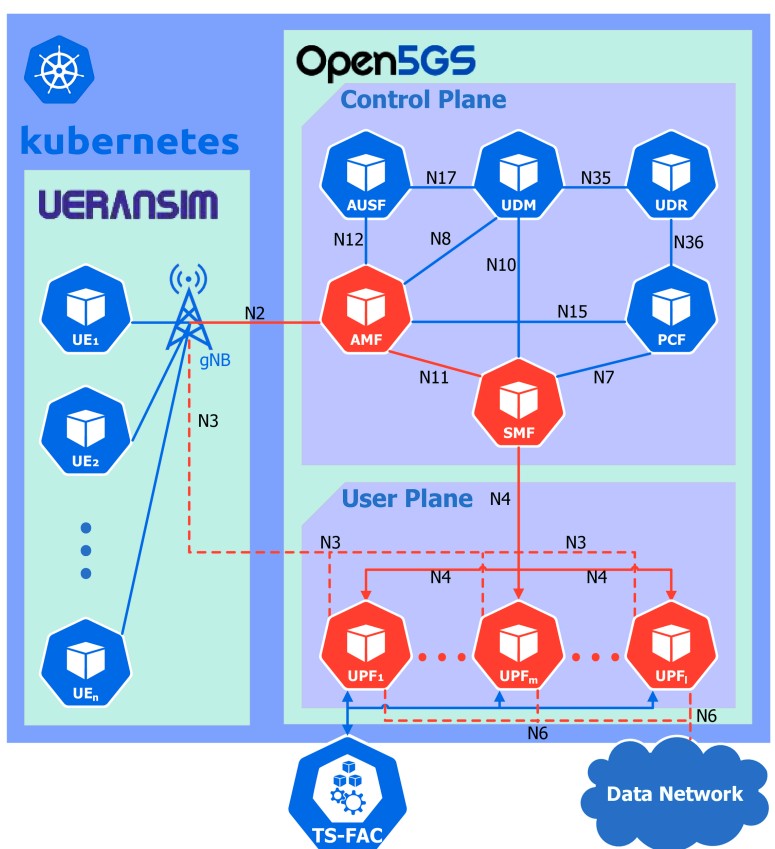

**Fig 9. Kubernetes-based Open5GS testbed implementing a complete 5G network infrastructure.** The red entities and lines represent the process of establishing the PDU session in the 5G core network.

In our experimental simulation model, Open5GS nodes have been deployed on Kubernetes, one of the most widely used container orchestration systems [30]. Accordingly, the functions of the 5G network, including UPF instances, are virtualized using container technology as illustrated in Fig 9. Core elements, including UPF instances of the 5G core network, are packaged into separate containers and organized into Pods within Kubernetes with resources and execution of each Pod managed by Kubernetes. This setup facilitates isolation of the execution environment and optimizes resource utilization. Kubernetes manages these Pods as the fundamental unit of deployment. In this paper, we consider a Pod running a single container that hosts an UPF image; thus, it can be referred to as an UPF Pod in Kubernetes [31]. We also use Open5GS [32] deployed on Kubernetes to simulate **Algorithm TS-FAC** and compare the results with the analytical outcomes [33]. A notable point is that in [2], the authors only performance with numerical evaluation. Therefore, to verify the accuracy of the improved model, we conducted additional simulations with a duration of 10,000 seconds. and due to the Poisson arrival process and the exponential service process, the results of each run exhibit slight but not significant variations. Therefore, to confirm accuracy, we conducted 100 to 300 simulations and compared the averaged results, as shown in Figs 10 and 11. From the average results presented in Figs 10 and 11, it is evident that the simulation results are accurate to 99% and exhibit convergence. Furthermore, to validate the accuracy of our theoretical model in alignment with practical simulations, the results are presented in Figs 20 and 21

## Results and discussions

Simulations are performed with the default parameters as shown in Table 2 (similar to [2]) to compare the performance of TSUPF-FAC and TSUPF in [2]. In some cases, the parameters are reset to suit the simulation objectives. In our model, it is assumed that the PDU session arrival rate follows a Poisson distribution, while the service rate follows an exponential distribution - an approach consistent with the assumptions in [2,12], among others. Furthermore, simulations have been carried out to verify the practical feasibility of the proposed model.

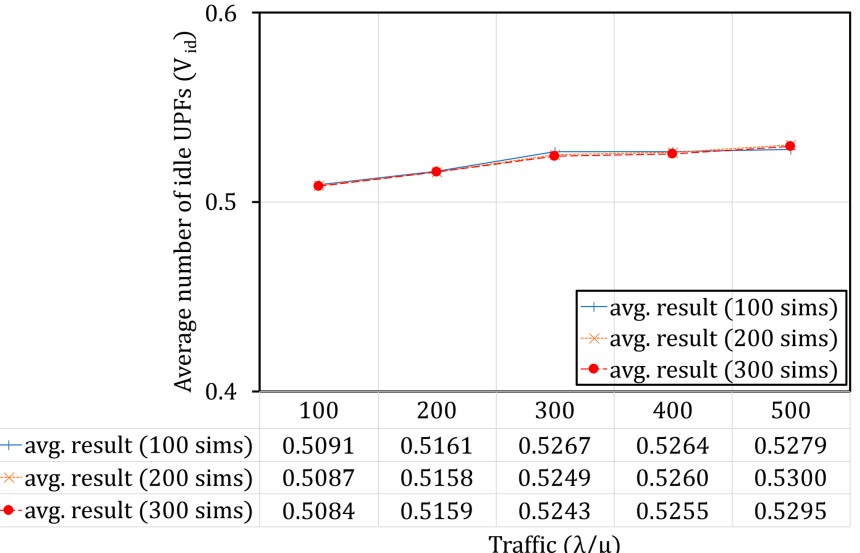

| | 100 | 200 | 300 | 400 | 500 |
|---|---|---|---|---|---|
| avg. result (100 sims) | 0.5091 | 0.5161 | 0.5267 | 0.5264 | 0.5279 |
| avg. result (200 sims) | 0.5087 | 0.5158 | 0.5249 | 0.5260 | 0.5300 |
| avg. result (300 sims) | 0.5084 | 0.5159 | 0.5243 | 0.5255 | 0.5295 |

Traffic ($\lambda/\mu$)

**Fig 10. Comparison results by number of simulations for $V_{id}$.**

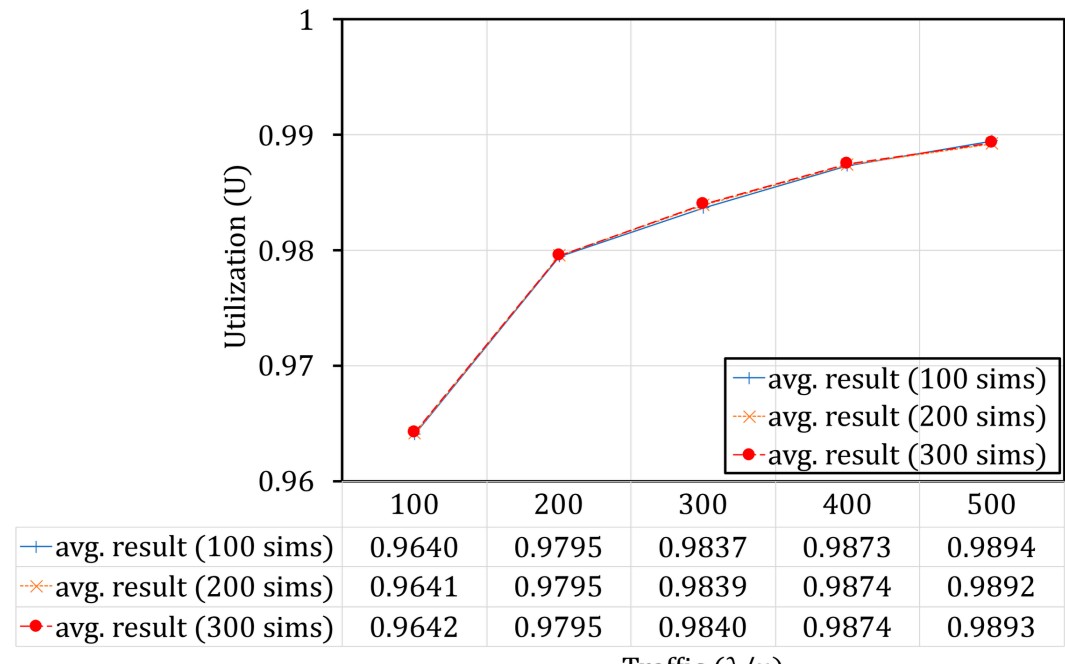

| | 100 | 200 | 300 | 400 | 500 |
|---|---|---|---|---|---|
| ┼ avg. result (100 sims) | 0.9640 | 0.9795 | 0.9837 | 0.9873 | 0.9894 |
| ⨯ avg. result (200 sims) | 0.9641 | 0.9795 | 0.9839 | 0.9874 | 0.9892 |
| ●  avg. result (300 sims) | 0.9642 | 0.9795 | 0.9840 | 0.9874 | 0.9893 |

Traffic ($\lambda/\mu$)

**Fig 11. Comparison results by number of simulations for *U*.**

**Table 2. Simulation/analysis parameters and values.**

| Notations | Value ranges | Default values |
|---|---|---|
| $C$ | 8 | 8 |
| $M$ | 1 | 1 |
| $L$ | 80 | 80 |
| $\lambda/\mu$ | $[100,500]$ | 500 |
| $T_1$ | $[1,7]$ | 2 |
| $T_2$ | $[11,17]$ | 13 |
| $H_1$ | $[\lceil|\mathcal{S}|/10\rceil,|\mathcal{S}|]$ | $|\mathcal{S}|/3$ |
| $H_2$ | $[\lceil|\mathcal{S}|/10\rceil,|\mathcal{S}|]$ | $2|\mathcal{S}|/3$ |
| $\alpha_1$ | $[0.6,1]$ | 0.95 |
| $\alpha_2$ | $[0.6,1]$ | 0.9 |

The performance evaluation metrics are as mentioned in the Sect 'Performance evaluation metrics':

- The average number of idle UPF instances (noted $V_{id}$): results are computed using (12) according to the theory and are averaged from 100 to 300 simulations, and
- The utilization (noted $U$): results are computed using (13) according to the theory and are averaged from 100 to 300 simulations.

According to the definition of the traffic load $\rho = \lambda/(k\mu)$ [34], where $k$ is the number of servers. Based on the parameter values in the paper, we deduce $k \leq C \times L = 640$. For the system to operate stably, we need the condition $\rho \leq 1$ or $\lambda/\mu \leq 640$. Therefore, we choose to simulate and analyze with the value $\lambda/\mu \leq 500$ is appropriately chosen to simulate and analyze.

In addition, assuming that there are multiple UE and each UE can request the establishment of multiple PDU sessions.

### Comparison of $V_{id}$ and $U$ when varying $H_1$ and $H_2$

The first goal of our simulation is to evaluate the effectiveness of integrating the FAC mechanism into Q-TSUPF. Figs 12 and 13 show that Q-TSUPF in [2] is represented by a large red circle marker corresponding to $H_1 = H_2 = |\mathcal{S}|$ or $H_1/|\mathcal{S}| = H_2/|\mathcal{S}| = 1$. We adjust the thresholds $H_1$ and $H_2$ relative to the total number of system states $|\mathcal{S}|$ (1).

The results show that when $H_2 \leq 7|\mathcal{S}|/10$, the number of idle UPF instances remains almost unchanged at around 0.45 (the lowest value). However, when $H_2 > 7|\mathcal{S}|/10$, the number of idle UPF instances gradually changes according to the value of $H_2$, with the exception of the case where $H_1 = H_2 = |\mathcal{S}|$, which corresponds to the Q-TSUPF model in [2], yielding the highest number of idle UPF instances. In terms of performance, the results remain stable at a level above 0.9921, or 99,21%, which demonstrates the suitability of our proposed theoretical model. As shown in Fig 12, the number of idle UPF instances is lower, but the utilization level does not change. The result suggests that integrating FAC into Q-TSUPF makes the number of UPF instances deployed more efficiently while maintaining the same utilization level.

### Comparison of $V_{id}$ and $U$ when varying $\alpha_1$ and $\alpha_2$

In the above section, we consider scaling a UPF instance once the number of served PDU sessions reaches the threshold $H_1$ and $H_2$. However, to create a smooth and non-abrupt transition, two control probabilities $\alpha_1$ and $\alpha_2$ are added for the transition at $H_1$ and $H_2$, respectively. Assuming the threshold values are fixed as $H_1 = (1/3)|\mathcal{S}|$ and $H_2 = (2/3)|\mathcal{S}|$, Fig 14 depicts $V_{id}$ as $\alpha_1$ and $\alpha_2$ vary, where $V_{id}$ of Q-TSUPF-FAC is lower than that of Q-TSUPF [2]. Q-TSUPF [2] is indicated by a large red circle marker corresponding to $\alpha_1 = \alpha_2 = 1$. Despite reducing $V_{id}$, $U$ of Q-TSUPF-FAC is not lower than Q-TSUPF [2]. As shown in Fig 15,

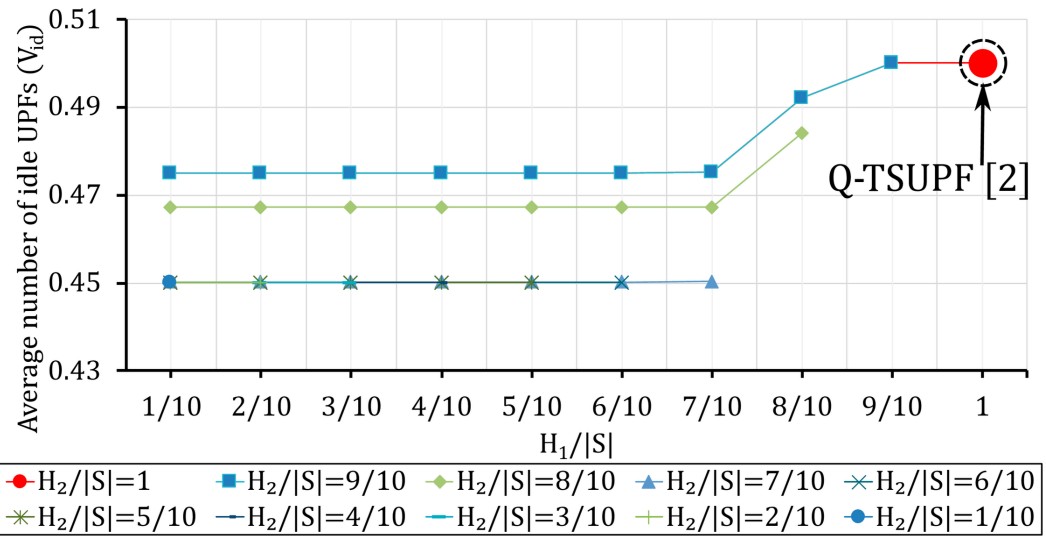

**Fig 12. Comparison of the performance of Q-TSUPF-FAC and Q-TSUPF [2] based on $V_{id}$ when varing $H_1$ and $H_2$.** The large red marker represents the special case of Q-TSUPF [2].

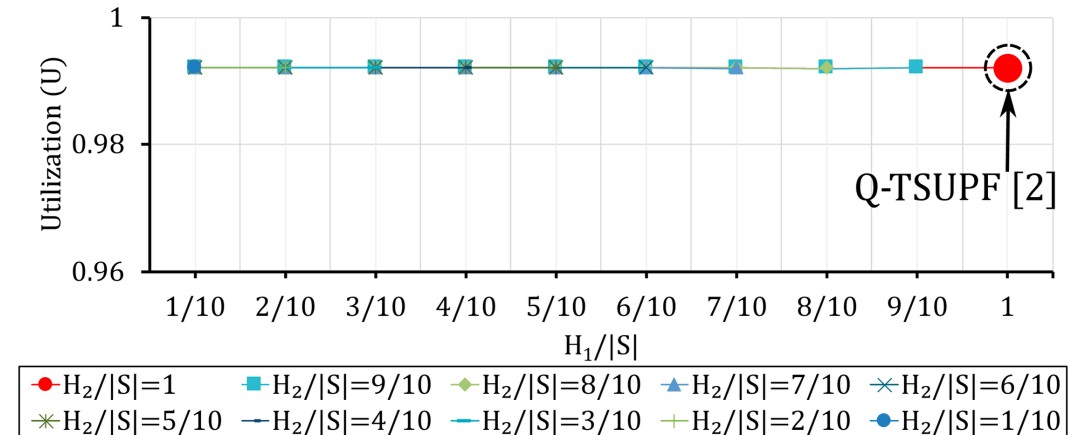

**Fig 13. Comparison of the performance of Q-TSUPF-FAC and Q-TSUPF [2] based on $U$ when varing $H_1$ and $H_2$.** The large red marker represents the special case of Q-TSUPF [2].

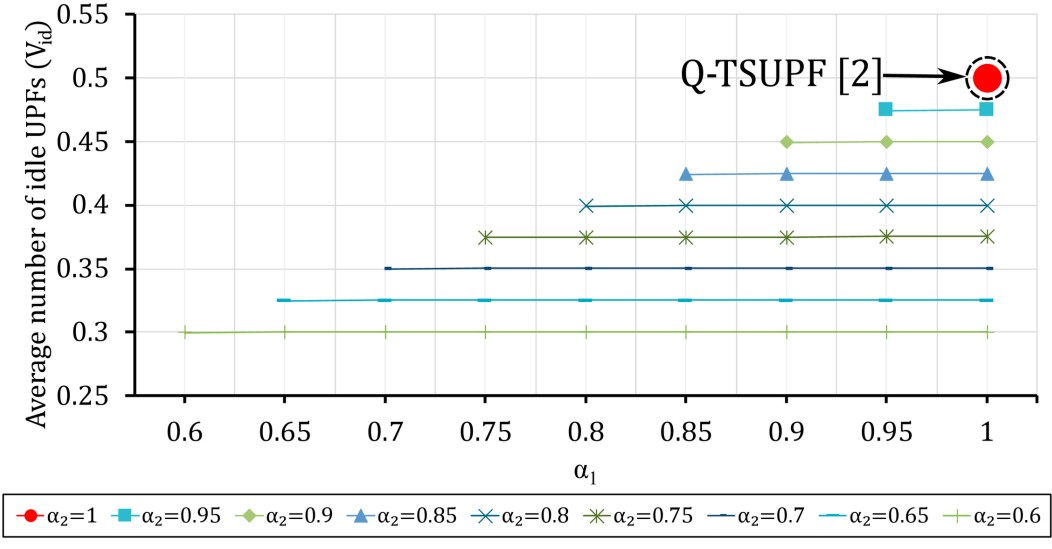

**Fig 14. Comparison of the performance of Q-TSUPF-FAC and Q-TSUPF [2] based on $V_{id}$ when varing $\alpha_1$ and $\alpha_2$.** The large red marker represents the special case of Q-TSUPF [2].

Q-TSUPF-FAC always maintains a stable efficiency level at 0.9921, similar to that of Q-TSUPF [2].

### Comparison of $V_{id}$ and $U$ when varying $T_1$ and $T_2$

In [2], the system performance was analyzed with different thresholds $T_1$ and $T_2$. As shown in Figs 16 and 17, with the predefined values of $(H_1, H_2, \alpha_1, \alpha_2)$ as in Table 2 and different considered pairs of $(T_1, T_2)$, $V_{id}$ of Q-TSUPF-FAC are always lower than Q-TSUPF in [2] (Fig 16). However, $U$ of Q-TSUPF-FAC achieves a higher value than Q-TSUPF [2] (Fig 17), which again proves the effectiveness of the FAC mechanism. Although the results in Fig 17 show that the TS-FAC algorithm only increases the performance by about 1%, the improvement is noticeable as each UPF can serve more PDU sessions.

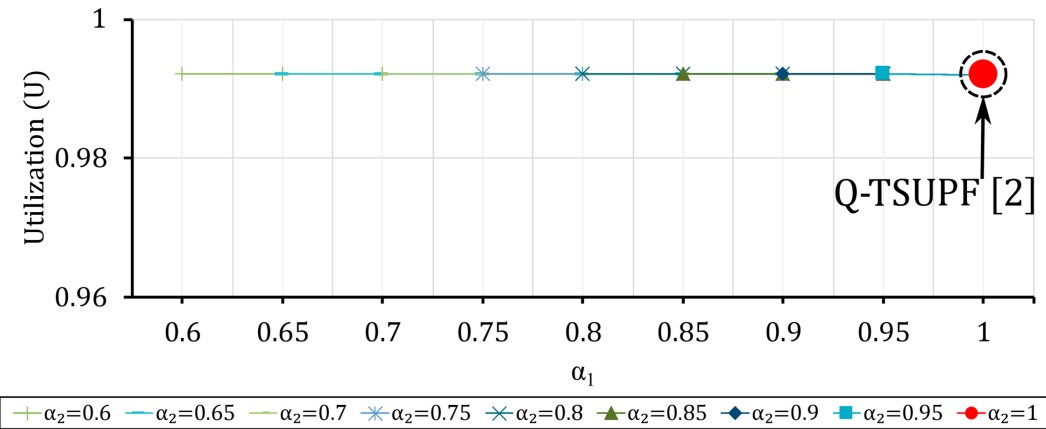

**Fig 15. Comparison of of the performance of Q-TSUPF-FAC and Q-TSUPF [2] based on $U$ when varing $\alpha_1$ and $\alpha_2$.** The large red marker represents the special case of Q-TSUPF [2].

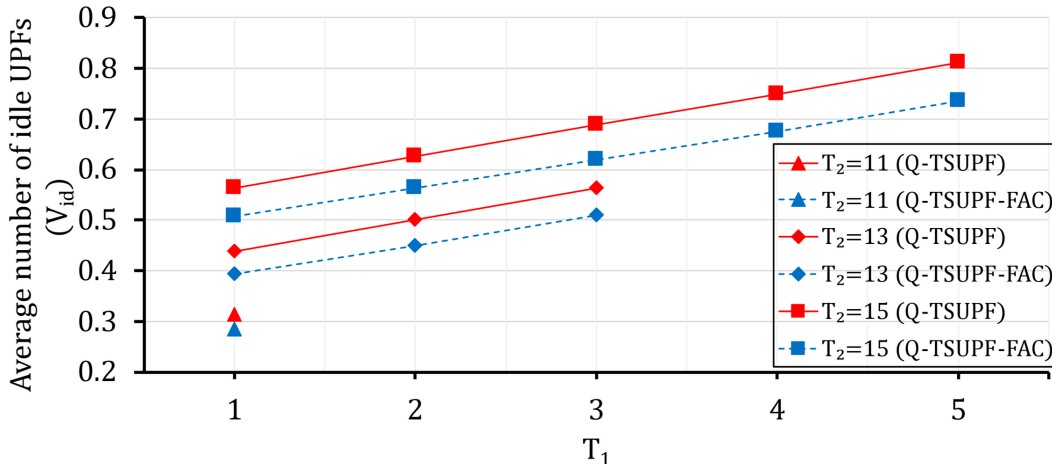

**Fig 16. Comparison of the performance of Q-TSUPF-FAC and Q-TSUPF [2] based on $V_{id}$ when varing $T_1$ and $T_2$.**

### Comparison of $V_{id}$ and $U$ when varying traffic $\lambda/\mu$

The efficiency of scaling UPF instances is affected by traffic $\lambda/\mu$. Specifically, when changing intense (from 100 to 500), Fig 18 shows an increase in $V_{id}$. An increase is necessary to respond quickly to requests when the number of arriving PDU sessions increases and thus maintain the stable utilization of the system. Fig 19 shows the resource utilization adapted to the increasing arrival rate of these PDU sessions.

### Comparison of $V_{id}$ and $U$ between theorical analysis and simulation

In order to demonstrate the correctness of the theoretical model and simulation implementation, we compare the theoretical analysis results and simulation results with the defaultparameters, as shown in Table 2. Figs 20 and 21 show that there is a similarity of the

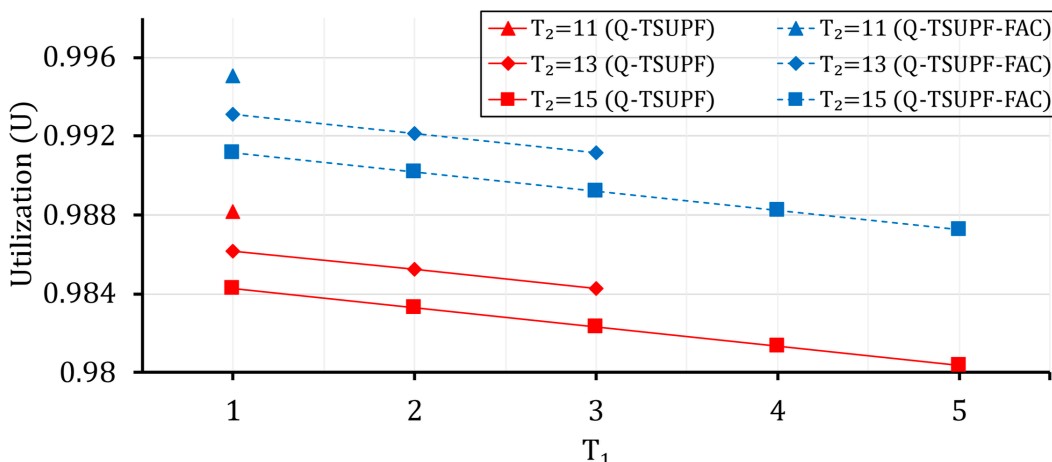

**Fig 17. Comparison of of the performance of Q-TSUPF-FAC and Q-TSUPF [2] based on $U$ when varing $T_1$ and $T_2$.**

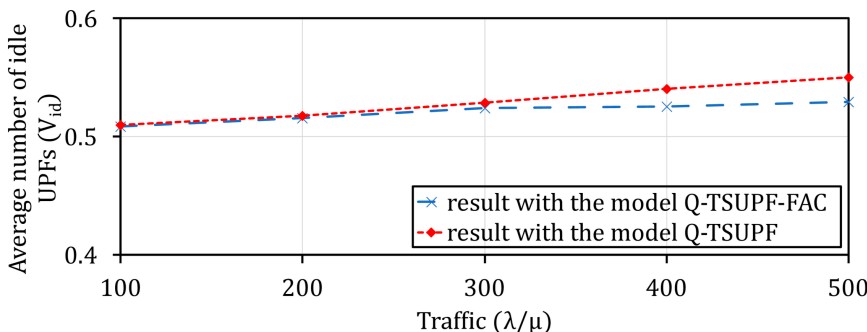

**Fig 18. Comparison of the performance of Q-TSUPF-FAC and Q-TSUPF [2] based on $V_{id}$ when varing $\lambda/\mu$.**

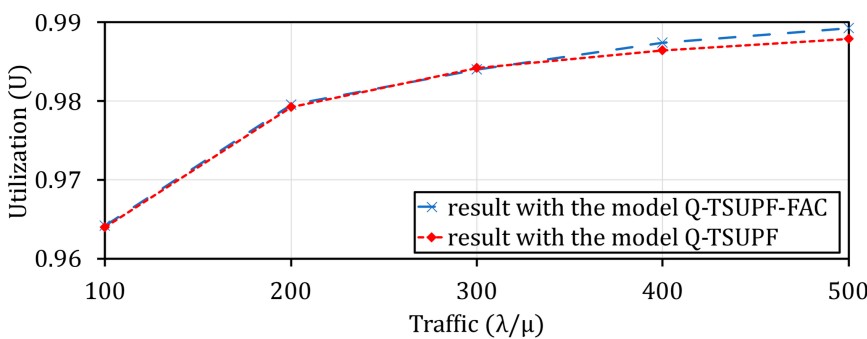

**Fig 19. Comparison of the performance of Q-TSUPF-FAC and Q-TSUPF [2] based on $U$ when varing $\lambda/\mu$.**

performance curves ($V_{id}$ and $U$) between the theoretical analysis and the experimental simulation. It is clear that adding FAC to the Q-TSUPF model [2] is correct and brings better performance for scaling UPF instances in 5G core systems.

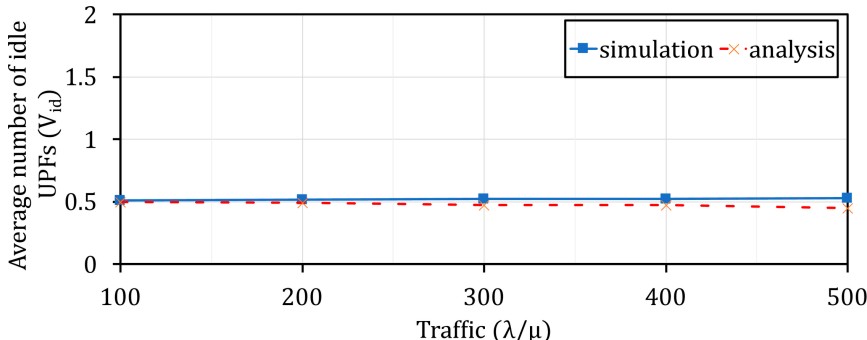

**Fig 20. Comparison of $V_{id}$ between theorical analysis and simulation when varying $\lambda/\mu$.**

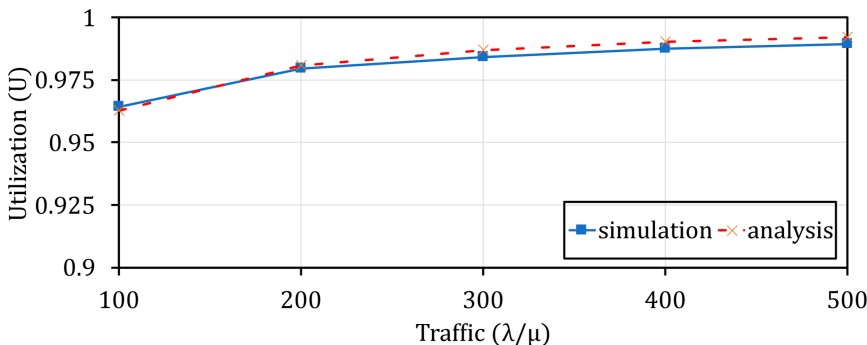

**Fig 21. Comparison of $U$ between theorical analysis and simulation when varying $\lambda/\mu$.**

## Conclusion

In this paper, we have established a threshold scaling and controlling model by proposing and integrating the FAC mechanism to threshold scaling model (Q-TSUPF). Additionally, we have developed an effective algorithm for applying the FAC mechanism, which aids in the analysis, calculation, and real-time simulation of the queueing model. Simulation and analytical results indicate that our model outperforms existing models by incorporating the control thresholds $H_1$ and $H_2$ with the control probabilities $\alpha_1$ and $\alpha_2$, respectively. Our model results in a lower number of idle UPF instances compared to the scenario without the FAC mechanism under system traffic load, thus allowing the system to conserve distributed resources while maintaining high performance relative to incoming request volumes. A notable advantage is the ease of implementing this FAC mechanism. The computational complexity of the algorithm is $O(C \times L)$, but it is more efficient regarding resource utilization in fewer idle UPF instances. Furthermore, it enables network operators to evaluate and ensure that a 5G network model can meet QoS requirements for PDU session requests. A crucial aspect is that our model can be applied in 6G networks. To fulfill the requirements in heterogeneous environments, we will consider applying random processes other than Poisson processes. Additionally, artificial intelligence and machine learning approaches will also be considered to improve computational results in heterogeneous networks.

## Author contributions

**Conceptualization:** Thanh Chuong Dang.

**Data curation:** Ly Cuong Hoa.

**Formal analysis:** Ly Cuong Hoa.

**Methodology:** Thanh Chuong Dang.

**Project administration:** Thanh Chuong Dang.

**Software:** Ly Cuong Hoa.

**Supervision:** Viet Minh Nhat Vo.

**Writing – original draft:** Ly Cuong Hoa, Thanh Chuong Dang.

**Writing – review & editing:** Thanh Chuong Dang, Viet Minh Nhat Vo.

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
