## [Decision Letter · Decision Letter 0]

26 Nov 2024

PONE-D-24-41605An Integrated Model of Threshold-based Scaling and Fractional Admission Controlling to Improve Resource Utilization Efficiency in 5G Core NetworksPLOS ONE

Dear Dr. Dang,

Thank you for submitting your manuscript to PLOS ONE. After careful consideration, we feel that it has merit but does not fully meet PLOS ONE’s publication criteria as it currently stands. Therefore, we invite you to submit a revised version of the manuscript that addresses the points raised during the review process.

We look forward to receiving your revised manuscript.

Kind regards,

Dhanamjayulu C, Ph.D & Post.Doc

Academic Editor

PLOS ONE

Journal Requirements:

2. Please note that PLOS ONE has specific guidelines on code sharing for submissions in which author-generated code underpins the findings in the manuscript. In these cases, we expect all author-generated code to be made available without restrictions upon publication of the work. 

Please review our guidelines at https://journals.plos.org/plosone/s/materials-and-software-sharing#loc-sharing-code and ensure that your code is shared in a way that follows best practice and facilitates reproducibility and reuse.

“This work was supported by the Ministry of Education and Training (Vietnam) for the 362 development of Science and Technology under grant number B2023-DHH-17.”

4. We note that your Data Availability Statement is currently as follows: 

“All relevant data are within the manuscript and its Supporting Information files.”

**Additional Editor Comments:**

The reviewers recommend reconsideration the manuscript with revision and modification. I invite the authors to resubmit the manuscript after addressing the comments raised by the reviewers.

Reviewers' comments:

Reviewer's Responses to Questions

**Comments to the Author**

1. Is the manuscript technically sound, and do the data support the conclusions?

Reviewer #1: Yes

Reviewer #2: Yes

2. Has the statistical analysis been performed appropriately and rigorously? 

Reviewer #1: Yes

Reviewer #2: Yes

3. Have the authors made all data underlying the findings in their manuscript fully available?

Reviewer #1: No

Reviewer #2: Yes

4. Is the manuscript presented in an intelligible fashion and written in standard English?

Reviewer #1: Yes

Reviewer #2: Yes

5. Review Comments to the Author

Reviewer #1: The manuscript successfully addresses resource optimization in 5G core networks by integrating threshold-based scaling and fractional admission control. The methodology is sound, and the results are validated through theoretical and experimental alignment. For improvement, I suggest the following:

1. Explicitly address how this model can adapt to varying traffic loads and user behaviors in heterogeneous environments.

2. There is no explicit mention of data repositories or links for external data availability, which may be necessary for full compliance with PLOS data policies. Then you have to include raw simulation data or repository links for transparency and replication.

3. Review for minor grammatical errors and consistency in technical terminology. For example, standardize terms like "UPF instance" and "Pod" where necessary.

4. Enhancing figure captions with detailed descriptions would improve their standalone interpretability. So add Figures' captions.

Reviewer #2: I have reviewed the manuscript titled "An Integrated Model of Threshold-based Scaling and Fractional Admission Controlling to Improve Resource Utilization Efficiency in 5G Core Networks." The study presents a novel integration of fractional admission control (FAC) and threshold-based scaling to optimize resource utilization in 5G core networks. By implementing the proposed TSUPF-FAC algorithm and validating it using Kubernetes-based Open5GS, the paper demonstrates significant improvements in resource efficiency and system performance. Below, I offer feedback for further improvement:

Content and Structure: The manuscript is well-structured, with a clear presentation of the problem, methodology, and experimental evaluation. However, the introduction could benefit from additional context on the broader implications of resource optimization in 5G networks, such as its impact on service quality and cost reduction for service providers. Expanding the discussion section to include potential scalability challenges and real-world deployment scenarios would also strengthen the manuscript.

Literature Review and Citations: The literature review includes relevant references but could be further enriched with recent studies on optimization and machine learning in resource management for 5G networks. I recommend incorporating the following references to align the manuscript with current advancements in the field:

https://doi.org/10.1016/j.eswa.2023.122147

https://doi.org/10.21608/jaiep.2024.386693

https://doi.org/10.54216/JAIM.080103

These citations will enhance the manuscript's contextual relevance and support the discussion on optimization and system performance.

Technical Clarifications and Suggestions: While the proposed model and results are promising, additional details in the following areas would improve clarity and reproducibility:

FAC Threshold Selection: Elaborate on the criteria for selecting H1 and H2 thresholds and their impact on system performance.

Computational Efficiency: Discuss the computational complexity of the TSUPF-FAC algorithm, particularly in large-scale deployments with high traffic loads.

Real-World Applicability: Provide insights into how the proposed model could be integrated into existing 5G networks and its adaptability to future 6G architectures.

Comparative Analysis: Expand the discussion on how the proposed model outperforms other state-of-the-art scaling and admission control techniques in terms of efficiency and scalability.

I hope these suggestions assist in refining your manuscript. The integration of FAC with threshold-based scaling represents a significant advancement in resource optimization for 5G core networks, and with these revisions, the study could make a substantial contribution to the field of network management and optimization.

6. PLOS authors have the option to publish the peer review history of their article (what does this mean?). If published, this will include your full peer review and any attached files.

Reviewer #1: No

Reviewer #2: No

---

## [Author Response · Author response to Decision Letter 1]

11 Dec 2024

We would like to update the content of our Funding Statement in the online submission system to read: “This work was supported by the Ministry of Education and Training (Vietnam) for the development of Science and Technology under grant number B2023-DHH-17.” Please update the online submission form on our behalf.

---

## [Decision Letter · Decision Letter 1]

1 Jan 2025

PONE-D-24-41605R1An Integrated Model of Threshold-based Scaling and Fractional Admission Controlling to Improve Resource Utilization Efficiency in 5G Core NetworksPLOS ONE

Dear Dr. Dang,

Thank you for submitting your manuscript to PLOS ONE. After careful consideration, we feel that it has merit but does not fully meet PLOS ONE’s publication criteria as it currently stands. Therefore, we invite you to submit a revised version of the manuscript that addresses the points raised during the review process.

**ACADEMIC EDITOR: **The reviewers recommend reconsideration the manuscript with revision and modification. I invite the authors to resubmit the manuscript after addressing the comments raised by the reviewers.

We look forward to receiving your revised manuscript.

Kind regards,

Dhanamjayulu C, Ph.D & Post.Doc

Academic Editor

PLOS ONE

Journal Requirements:

Additional Editor Comment:

The reviewers recommend reconsideration the manuscript with revision and modification. I invite the authors to resubmit the manuscript after addressing the comments raised by the reviewers.

Reviewers' comments:

Reviewer's Responses to Questions

**Comments to the Author**

1. If the authors have adequately addressed your comments raised in a previous round of review and you feel that this manuscript is now acceptable for publication, you may indicate that here to bypass the “Comments to the Author” section, enter your conflict of interest statement in the “Confidential to Editor” section, and submit your "Accept" recommendation.

Reviewer #1: (No Response)

Reviewer #2: All comments have been addressed

2. Is the manuscript technically sound, and do the data support the conclusions?

Reviewer #1: Yes

Reviewer #2: Yes

3. Has the statistical analysis been performed appropriately and rigorously? 

Reviewer #1: Yes

Reviewer #2: Yes

4. Have the authors made all data underlying the findings in their manuscript fully available?

Reviewer #1: Yes

Reviewer #2: Yes

5. Is the manuscript presented in an intelligible fashion and written in standard English?

Reviewer #1: Yes

Reviewer #2: Yes

6. Review Comments to the Author

Reviewer #1: While the manuscript mentions some practical considerations for real-world applications, a more detailed discussion of potential limitations (e.g., impact of network dynamics on performance) would strengthen the paper and provide insights into areas for future improvement.

Reviewer #2: (No Response)

7. PLOS authors have the option to publish the peer review history of their article (what does this mean?). If published, this will include your full peer review and any attached files.

Reviewer #1: No

Reviewer #2: **Yes: **Abdelaziz Abdelmoniem Abdelhamid

---

## [Author Response · Author response to Decision Letter 2]

7 Jan 2025

To the Editor: We have reviewed and revised according to the journal’s format and named the files as requested.

- Response to Reviewers.

- Revised Manuscript with Track Changes.

- Manuscript.

- Cover Letter.

To the Trviewer #1:

In our model, we assume a static network environment with arrival rates and service rates for PDU session requests, which is a common simplification in theoretical studies, and we have provided parameter values taken from [2]. However, we acknowledge that network dynamics in real-world systems can significantly impact the performance of the model, and we aim to provide a more detailed discussion of this issue.

Impact of Network Dynamics on Performance

1. Fluctuations in traffic patterns: In real-world networks, traffic patterns are highly variable and influenced by factors such as user mobility, application demand, and network congestion. The arrival rates of PDU session requests may not always follow a Poisson distribution, as assumed in our model. Instead, traffic may exhibit burstiness, long-range dependence, or periods of heavy congestion that can lead to queueing delays and packet losses. These fluctuations can significantly degrade the performance of the network, affecting key metrics such as throughput, latency, and packet drop rates.

2. Heterogeneity in network resources: Networks are often heterogeneous, with varying capacities across different access technologies (e.g., LTE, 5G, Wi-Fi) and network infrastructure (e.g., core vs. edge network). The performance of our model may be slightly influenced if these heterogeneous resources are not considered, particularly regarding the allocation and management of different network slices or frequency bands. In practical scenarios, network performance may vary depending on load balancing, resource contention, and interference between different network segments, which could affect the QoS during the establishment and maintenance of PDU sessions.

3. Network congestion and load balancing: Dynamic congestion, where certain parts of the network experience higher traffic volumes than others, can influence the ability of the network to establish or maintain PDU sessions. Load balancing algorithms must continuously adjust to the network’s changing conditions to avoid bottlenecks. In real-time, if the algorithm fails to adapt effectively to congestion spikes or traffic shifts, session establishment delays or session failures could occur, leading to reduced network efficiency and QoS degradation. Our algorithm TSFAC has accounted for idle resources to ensure that a sudden surge in incoming traffic does not cause congestion.

4. Mobility and user behavior: In mobile networks, user mobility is another significant factor. Users constantly move between cells or access networks, and the network must adapt to this mobility to maintain stable connections. A model that assumes static conditions might not capture the effects of handovers, changing signal strengths, or cell overloads. These factors can impact the time taken to establish PDU sessions, the continuity of sessions, and even session reliability, especially in high-mobility scenarios like vehicles or public events.

5. Environmental interference and external factors: In many practical settings, external interference (such as radio frequency interference, weather conditions, or physical obstructions) can also affect the performance of the network. These dynamic factors can cause fluctuations in network availability and quality, leading to disruptions in service that could affect session establishment and resource allocation.

Addressing Network Dynamics in Future Work

To better capture these real-world challenges, we plan to extend our model by incorporating non-Poisson traffic models, such as the General distribution (G) or Erlang distribution (Ek), to simulate bursty traffic and congestion scenarios more accurately. Furthermore, we will explore adaptive load balancing mechanisms and mobility models that account for the dynamic behavior of users and network resources in heterogeneous environments. This will allow us to evaluate the model’s resilience and performance under varying levels of network load and mobility.

Additionally, as previously mentioned, we will incorporate machine learning (ML) techniques to predict and adjust to network dynamics in real-time. ML can help model the time-varying nature of traffic and network conditions, enabling dynamic optimization of resources, congestion management, and traffic prediction. This approach can also help us automatically tune the parameters of the model based on real-time data, improving its adaptability to network fluctuations in practical scenarios.

We have updated at the lines from 371 to 374: “To fulfill the requirements in heterogeneous environments, we will consider applying random processes other than Poisson processes. Additionally, artificial intelligence and machine learning approaches will also be considered to improve computational results in heterogeneous networks.”.

---

## [Decision Letter · Decision Letter 2]

3 Jun 2025

PONE-D-24-41605R2An Integrated Model of Threshold-based Scaling and Fractional Admission Controlling to Improve Resource Utilization Efficiency in 5G Core NetworksPLOS ONE

Dear Dr. Dang,

Thank you for submitting your manuscript to PLOS ONE. After careful consideration, we feel that it has merit but does not fully meet PLOS ONE’s publication criteria as it currently stands. Therefore, we invite you to submit a revised version of the manuscript that addresses the points raised during the review process.

**ACADEMIC EDITOR: **The reviewers recommend reconsideration the manuscript with revision and modification. I invite the authors to resubmit the manuscript after addressing the comments raised by the reviewers.

We look forward to receiving your revised manuscript.

Kind regards,

Dhanamjayulu C, Ph.D & Post.Doc

Academic Editor

PLOS ONE

Journal Requirements:

Additional Editor Comments:

The reviewers recommend reconsideration the manuscript with revision and modification. I invite the authors to resubmit the manuscript after addressing the comments raised by the reviewers.

Reviewers' comments:

Reviewer's Responses to Questions

**Comments to the Author**

1. If the authors have adequately addressed your comments raised in a previous round of review and you feel that this manuscript is now acceptable for publication, you may indicate that here to bypass the “Comments to the Author” section, enter your conflict of interest statement in the “Confidential to Editor” section, and submit your "Accept" recommendation.

Reviewer #3: (No Response)

Reviewer #4: All comments have been addressed

2. Is the manuscript technically sound, and do the data support the conclusions?

Reviewer #3: Partly

Reviewer #4: Yes

3. Has the statistical analysis been performed appropriately and rigorously? 

Reviewer #3: Yes

Reviewer #4: Yes

4. Have the authors made all data underlying the findings in their manuscript fully available?

Reviewer #3: Yes

Reviewer #4: (No Response)

5. Is the manuscript presented in an intelligible fashion and written in standard English?

Reviewer #3: Yes

Reviewer #4: Yes

6. Review Comments to the Author

Reviewer #3: On lines 33 – 140 there is information about the use of a model based on a Poission distribution of session arrivals. This has already been risen in the previous comments by other reviewers, but I think the response added to the paper is not sufficient. The references to the literature or an analysis should be provided showing if the session arrivals in 5G have a Poisson distribution or how far from the Poisson distribution they are.

The figures 5 and 6 don’t add much to the paper. They show that the simulation results are consistent for different seeds, which is good, but this could be shown more clearly by adding the confidence intervals on the plots or listing them in the table in the paper. Instead of comparing the simulations to simulations, the model should be validated by comparison of results from Open5GS testbed with the results obtained using Markov chain model. Has such validation for the Q-TSUPF-FAC been made?

It would be more clear to use the name of the metric in the figures caption, namely the utilization of radio resources instead just U, or the average number of idle instances instead of Vid.

The figures 7 and 8 are supposed to compare the Q-TSUPF-FAC and Q-TSUPF algorithms, but it’s very hard to understand from the plots what is the difference. Especially on figure 8, the metric U remains unchanged, regarding the parameters used. Does it mean that the utilization remains the same for both algorithms? Why then the number of idle instances is different?

The gain on the better utilization of Q-TSUPF-FAC shown on the figure 12 is very small – it is smaller than 1 percent point. There is too little comment in the paper, why the authors propose to complicate the algorithm for such a small increase of efficiency?

The conclusions seems to not correctly reflect the content of the paper. It is stated in line 368 that the advantage of the algorithm is the ease of implementation. In my understanding, the proposed method is a slight modification of the previously developed by the same authors Q-TSUPF algorithm. The previous algorithm is less complicated and easier to implement, while the currently proposed Q-TSUPF-FAC offers a slightly better utilization of resources at the cost of using more threshold for the scaling of the load. This should be clearly stated in the conclusions. Additionally, the fact that the model can be applied in 6G networks is both true for the current algorithm and for the previous one.

There is no analysis of the probability of not allocating the sufficient number of UPF instances. I understand that this is caused by the simplification made in the model that the allocation time is equal 0. In such a case the optimal strategy is to allocate the UPF instance as late as possible. A trivial strategy to configure the two thresholds used in the Q-TSUPF algorithm to the maximum possible values should be optimal in such a scenario. Is this a case and why then complicate the algorithm instead of selecting the right thresholds for the already existing method?

Reviewer #4: The authors significantly address all the reviewers' comments, and this version of the manuscript is technically acceptable for publication.

7. PLOS authors have the option to publish the peer review history of their article (what does this mean?). If published, this will include your full peer review and any attached files.

Reviewer #3: No

Reviewer #4: No

---

## [Author Response · Author response to Decision Letter 3]

11 Jun 2025

We have reviewed and revised according to the journal’s format and named the files as requested.

- Response to Reviewers.

- Revised Manuscript with Track Changes.

- Manuscript.

- Cover Letter.

We would like to thank you for your critical comments

---

## [Decision Letter · Decision Letter 3]

27 Jul 2025

An Integrated Model of Threshold-based Scaling and Fractional Admission Controlling to Improve Resource Utilization Efficiency in 5G Core Networks

PONE-D-24-41605R3

Dear Dr. Chuong Thanh Dang,

We’re pleased to inform you that your manuscript has been judged scientifically suitable for publication and will be formally accepted for publication once it meets all outstanding technical requirements.

Kind regards,

Dhanamjayulu C, Ph.D & Post.Doc

Academic Editor

PLOS ONE

Additional Editor Comments (optional):

The authors have addressed the reviewers’ comments properly

The article can be accepted for the publication in present form

Reviewers' comments:

Reviewer's Responses to Questions

**Comments to the Author**

1. If the authors have adequately addressed your comments raised in a previous round of review and you feel that this manuscript is now acceptable for publication, you may indicate that here to bypass the “Comments to the Author” section, enter your conflict of interest statement in the “Confidential to Editor” section, and submit your "Accept" recommendation.

Reviewer #4: All comments have been addressed

Reviewer #5: All comments have been addressed

2. Is the manuscript technically sound, and do the data support the conclusions?

Reviewer #4: Yes

Reviewer #5: Yes

3. Has the statistical analysis been performed appropriately and rigorously? 

Reviewer #4: Yes

Reviewer #5: Yes

4. Have the authors made all data underlying the findings in their manuscript fully available?

Reviewer #4: Yes

Reviewer #5: Yes

5. Is the manuscript presented in an intelligible fashion and written in standard English?

Reviewer #4: Yes

Reviewer #5: Yes

6. Review Comments to the Author

Reviewer #4: The author significantly addresses all the reviewers comments. This revised version of the paper has now technically eligible for the publication.

Reviewer #5: The authors revised the manuscript as per the suggestions given by the reviewers and hence the manuscript can be accepted for publication, after english corrections

7. PLOS authors have the option to publish the peer review history of their article (what does this mean?). If published, this will include your full peer review and any attached files.

Reviewer #4: No

Reviewer #5: **Yes: **Dr.Suresh Muthusamy

---

## [Editor Report · Acceptance letter]

PONE-D-24-41605R3

PLOS ONE

Dear Dr. Dang,

I'm pleased to inform you that your manuscript has been deemed suitable for publication in PLOS ONE. Congratulations! Your manuscript is now being handed over to our production team.

Kind regards,

on behalf of

Dr. Dhanamjayulu C

Academic Editor

PLOS ONE